# Extrusion Modification of Wheat Bran and Its Effects on Structural and Rheological Properties of Wheat Flour Dough

**DOI:** 10.3390/foods12091813

**Published:** 2023-04-27

**Authors:** Ranran Li, Chenyang Wang, Yan Wang, Xuan Xie, Wenjie Sui, Rui Liu, Tao Wu, Min Zhang

**Affiliations:** 1State Key Laboratory of Food Nutrition and Safety, College of Food Science and Engineering, Tianjin University of Science & Technology, Tianjin 300457, China; li17862686224@126.com (R.L.); wcy2020d@163.com (C.W.); ywang@mail.tust.edu.cn (Y.W.); lr@tust.edu.cn (R.L.); wutaoxx@gmail.com (T.W.); 2College of Food Science and Bioengineering, Tianjin Agricultural University, Tianjin 300392, China

**Keywords:** extrusion, wheat bran, dough, structure, rheological properties

## Abstract

The study investigated the extrusion modification of wheat bran and its effects on structural and rheological properties of wheat flour dough. Extruded bran showed better solubility of dietary fiber and structural porosity, leading to higher hydration and swelling power. Addition of extruded bran to dough caused water redistribution as an intensive aggregation of bound water to gluten matrix with reduced mobility. The bran–gluten interaction influenced by water sequestering caused partial gluten dehydration and conversion of *β*-turn into *β*-sheet, which demonstrated the formation of a more polymerized and stable gluten network. Farinographic data confirmed the promotion of dough stability with extruded bran addition at lower gluten content, while viscoelastic data suggested improved dough elasticity at all gluten contents by increasing elastic moduli and decreasing loss tangent. This study would be useful for interpreting the modification effect and mechanism of extrusion on cereal brans and provide valuable guidance for applying it as an effective modification technology on the commercial production of cereal bran and its flour products.

## 1. Introduction

To satisfy the rising requirement for a healthy and low-calorie diet, the development of cereal products rich in dietary fiber has been an efficient means to improve consumers’ dietary patterns [1,2,3]. As a good source of dietary fiber, wheat bran is the major by-product in flour mill processing. Wheat bran consists of epidermis, peel, seed, nucellus layer and aleurone layer [3,4]. It is principally formed of the higher contents of cellulose, vitamins and minerals, more so than endosperm [5,6]. Prior research indicated that dietary fibers in wheat bran had a positive impact on diabetes and could prevent colon cancer, and reduce diverticulosis and the formation of gallstones [6,7]. However, most of the wheat bran produced today is generally underutilized as animal feed or discarded as waste agricultural residues. Its use in human food is limited [5]. This phenomenon should be mainly owing to the high content of insoluble dietary fiber resulting in the deterioration of products’ quality, which is correlated with their structural and rheological properties. Wheat bran could detrimentally affect the gluten network, reduce the dough resilience and weaken the framework of gas cells, and thus, gas retention. These effects inevitably lead to inferior baking quality, such as leavening restriction, loaf volume reduction, gritty texture and undesirable mouthfeel and taste [8]. Therefore, it is quite necessary to understand the basis for negative effects of dietary fibers on bread quality, so as to develop efficient strategies to eliminate quality defects, especially those correlated to dough rheology and bread volume [9,10,11].

Extrusion is a high-temperature/pressure short-time process for the modification and production of various kinds of food products. In the continuous screw shearing and heating stage, material is gradually exposed to a large pressure at the die head, which lasts for about 30~300 s and the heating temperature is in the range of 50~180 °C [12]. A number of chemical changes occur in this stage: starch gelatinization, protein cross-linking and flavor generation, etc. [9,13]. When the material is extruded out of the machine head, the instantaneous decompression causes water flashing and steam expansion, resulting in a porous structure formed of product. Extrusion has certain advantages, including the reduction of microbial content, inactivation of some enzymes and anti-nutritional factors and modification of sensory characteristics [1,2,9]. As a multi-stage, multi-functional and thermal/mechanical process, it has permitted a variety of food applications [12,14]. Previous studies showed that extrusion could change the physico-chemical properties of wheat bran and prompt the bioavailability and functionality of soluble dietary fiber (SDF) in bran materials [1,4,14]. However, the specific influencing mechanism of treated bran on the structure and rheology of wheat dough still remained to be further unveiled.

It is known that both hydration and polymerization of gluten protein are critical processes for dough formation [15,16,17]. During the hydration process, water acts as a solvent that solubilizes low molecular proteins and induces conformational changes in the hydrophobic interactions between bran and gluten protein. In the following, polymerization of gluten protein has an impact on the construction of tridimensional viscoelastic network, which would interfere with dough mobility [15,18], so the interaction between bran and gluten during hydration and polymerization are closely related to the extensibility and elasticity of dough and the textural features of terminal products. Until now, there are several hypotheses describing the impact of this interaction on dough quality, which was strongly associated with a combination of water binding and size depletion [19,20]. It was proposed that arabinoxylans in dietary fibers of bran tightly bound water in the dough so as to decrease the available water participating in the development of the gluten network [10,19,20]. The dilution of gluten, hindrance of the gluten network and piercing of gas cells by the addition of fiber particles contributed to the size depletion phenomena [10]. In addition, ferulic acids were supposed to narrow the development time, raise the breakage rate and stiffness of dough, with the interference mechanism tentatively attributed to its interference with the significant disulphide interchange reaction [3,10]. In the bakery industry, it is significant to understand the rheological changes in wheat dough during processing, owing to their close correlation with quality attributes of flour products. The reduction of dough elasticity by disturbing gluten network formation would lead to the interruption of gas cells, low volume of bread and poor quality of baking products [10,12,14]. Therefore, it is necessary to explore the interactions of extruded bran and gluten protein in the dough system and on these bases evaluate the influence of extruded bran on structural and rheological properties of wheat dough, for the aim of providing valuable guidance for applying extrusion as an effective modification technology on the commercial production of bran-containing flour products.

The novelty of this work is to reveal the interaction variations of extruded bran and gluten protein in dough systems with different gluten levels and further explore the effects of extruded bran on the structural and rheological properties of wheat dough. After extrusion modification, the composition, micro-structure and hydration characteristics of wheat bran were measured; then 0~25% extruded bran was added to dough with different gluten contents and the major changes of water states and gluten conformation of dough were analyzed; lastly, the farinographic and dynamic viscoelastic properties of dough were characterized, to reveal the possible physical mechanism of structure and rheology changes in dough system with extruded bran addition.

## 2. Materials and Methods

### 2.1. Materials

Wheat bran was purchased from Xiangyuan Jinxiangzhai Trading Co., Ltd., Shanxi Province, China. Raw material is in the form of thin slices with a diameter of 3~5 mm and a thickness of 0.5~1 mm. Different-gluten wheat flours were purchased from Binzhou Zhongyu Food Co., Ltd., Shandong Province, China. The protein content, wet gluten content and alveograph parameters of wheat flour are listed in Table 1.

### 2.2. Extrusion Modification

The extrusion of wheat bran was by a DS56-X twin-screw co-rotating, self-wiping extruder (Jinan Saixin Machinery Co., Jinan, China) with a ratio of length to diameter of 25 and screw speed up to 600 rpm. Then, 1 kg wheat bran was processed by the extruder each batch with a water content of 25 (w/w, %). The barrel temperature in I, II and III heating blocks was set to 100 °C, 145 °C and 170 °C, respectively, and the feeding speed and screw speed of 10 r/min and 100 r/min were selected based on preliminary study [21]. The extrudates were oven-dried at 50 °C overnight followed by being finely ground and passed through a 60-mesh screen (particle size <250 μm).

### 2.3. Compositional Determination of Wheat Bran

The TDF, SDF and IDF contents were determined by the AOAC 991.43 method. Protein, starch and lipid content were measured according to AOAC 920.87, 996.11 and 983.23 methods.

### 2.4. Microscopic Structure Characterization of Wheat Bran

The SEM photographs of bran samples were taken using Hitachi SU-1510 system (Hitachi, Pleasanton, CA, USA).

### 2.5. Hydration Property Determination of Wheat Bran

Water-holding capacity (WHC), oil-holding capacity (OHC) and water swelling capacity (WSC) were determined according to Sui et al. [3].

### 2.6. Preparation of Wheat Dough with Addition of Wheat Bran

Dough was prepared by mixing different-gluten flour and wheat bran (0%, 5%, 10%, 15%, 20%, 25%) with water until the consistency of the dough peaked in a Brabender farinograph using a 50 g mixing bowl. At the end of mixing, doughs were collected and then sealed with preservative film for around 0.5 h to wake up. The characterization of LF-NMR was conducted within 30~45 min.

### 2.7. Fourier-Transformed Infrared Spectroscopy (FT-IR) of Wheat Dough

The protein secondary structure of dough samples was characterized by FTIR spectroscopy (IS50, Nicolet, Thermo Fisher, Waltham, MA, USA). For each measurement, the freeze-dried dough was mixed with KBr (1:150, *w*/*w*). Then the mixture was ground in an agate mortar to a fine powder. The powder was pressed into a plate and the FTIR spectra were recorded by 40 scans in the range of 400 to 4000 cm^−1^ with a resolution of 4 cm^−1^ [17].

### 2.8. Low-Field Nuclear Magnetic Resonance (LF-NMR) of Wheat Dough

The quantitative determination of water structure in dough using LF-NMR was carried out with a 22.4 MHz NMR Analyzer PQ001 (Niumag Co., Ltd., Shanghai, China). The 90° and 180° pulse length was firstly measured to obtain the spin–spin relaxation time (T_2_) by a CPMG pulse sequence. A total of 1000 echoes were collected. The pulse interval was 0.08 ms, the cycle delay was 1.5 s and 16 scans were collected.

### 2.9. Farinographic Property Determination of Wheat Dough

This property was determined according to the AACC Method 54-21 (2000) by a farinograph (Brabender Technologie GmbH & Co. KG, Duisburg, Germany). Farinographic properties, including water absorption, development time, stability time and degree of softening were recorded.

### 2.10. Dynamic Viscoelastic Property Determination of Wheat Dough

A quantity of 50 g dough mixed with or without wheat bran was kneaded by the Doughlab system (Perten Instruments, Hägersten, Sweden) at 65 r/min for 10 min. Afterwards, the mixture was maintained at 25 °C for 0.5 h to obtain test samples. Dynamic viscoelastic properties of dough with wheat bran were measured by a HAAKE MARS-III rheometer (Thermo Fisher Scientific Inc., Sunny-vale, CA, USA). A frequency sweep test on the dough was performed using a steel plate with a diameter of 20 mm gapped by 1 mm.

### 2.11. Statistical Analysis

Unless otherwise specified, 3 independent trials were conducted, and values are presented as the arithmetic means ± standard deviation (SD). Statistical differences were evaluated using ANOVA and Duncan’s multiple range test. Values were considered to be significantly different at *p* < 0.05 (SPSS for Window 18.0).

## 3. Results and Discussion

### 3.1. Physicochemical Properties of Extruded Wheat Bran

#### 3.1.1. Component Analysis

The basic components of extruded wheat bran are shown in Figure 1a. Protein, starch, lipids, TDF, SDF and IDF contents of raw wheat bran were 18.37 ± 0.12%, 33.38 ± 1.37%, 4.08 ± 0.24%, 41.29 ± 2.89%, 3.10 ± 0.22% and 38.28 ± 1.37%, respectively. After extrusion, there was no significant difference observed in TDF and protein contents. The yield of SDF was increased to 7.65%, and the yields of IDF and starch were decreased to 33.31% and 30.57%, respectively. The increased content of SDF and decreased content of IDF suggested the conversion of IDF to SDF by extrusion. This was due to the disruption of non-covalent or covalent bonds in IDF, resulting in the production of soluble small-molecular-weight compounds accompanied by the change of molecular polarity [12]. Our previous study found that the content of dietary fiber components, for instance, arabinoxylan and *β*-glucan, and the proportion of high molecular weight granules of SDF were generally increased during extrusion [20,21]. According to reports, the influence of extrusion on TDF content was less than that of boiling, autoclaving or steaming [22]. Some researchers proposed that the unchanged contents of TDF were mainly attributed to the simultaneous formation of resistant starch at the same time as degradation of dietary fiber [5], but since starch content was decreased (*p* ≤ 0.01), this was not the case in this study. The reason for the lowered starch content was the force degradation during extrusion, resulting in the cleavage of intermolecular hydrogen bonds and the degradation of macromolecules in starch. Additionally, there would be some hydration and gelatinization of the starch during extrusion [7,13,14]. The protein content of wheat bran after extrusion was almost unchanged, but extrusion has been reported to cause the hydration and denaturation of proteins, which in turn affected their biological activity or potency, such as the improvement of intrinsic digestibility, the reduction of trypsin inhibitor activity and drops in serum and liver cholesterol levels [7]. Therefore, extrusion treatment changed the main composition of wheat bran, greatly promoting the conversion of IDF to SDF.

#### 3.1.2. Swelling Capacity and Solvent Retention Capacity

Studies have reported that the water-related properties, oil extraction capacity and swelling capacity of dietary fiber materials in functional foods were closely related to the rheological and functional properties. WHC, OHC and WSC of wheat bran before and after extrusion are illustrated in Figure 1b. In particular, WHC improved from 1.79% to 2.22%, and WSC increased from 2.28% to 3.33% while OHC deteriorated by 11.19%. As illustrated above, the degradation of SDF and some materials with low polymerization degree and high water-binding ability would be prompted by extrusion. There was more exposure of hydrophilic groups, which improved the water-binding ability of wheat bran and thus increased the WHC and reduced the OHC. Moreover, the multiple forces including mixing, stirring, shearing and puffing in extrusion process led to a pronounced change in the three-dimensional structure of wheat bran. The produced porous and loose structure was proportional to the improved retention and swelling ability of water, thus leading to the increase of WHC and WSC. High WHC was positively correlated with good hydration properties. The high hydration performance could facilitate water absorption of dietary fibers and their expansion to a gelation form, thereby increasing the food viscosity and delaying or preventing the excess cholesterol absorption in food. Additionally, the excellent hydrophilic properties could promote bowel movements and reduce risks of constipation, diverticulosis and colorectal cancer [23,24,25].

#### 3.1.3. Morphological Characterization

The microstructures of unextruded and extruded wheat bran are displayed in Figure 2. The structure of the raw sample was dense. The surface was smooth and had no obvious pores, factures or faults. After extrusion, the surface of bran became loose, porous and rough, with some small holes and obvious fragments or faults on it. During the extrusion process of wheat bran, based on the principle of gas phase change and thermal pressing effect, the high shearing action between screw and barrel as well as the heating action outside the sleeve jointly produced the internal pressure in the barrel. Under such conditions of high temperature, pressure and shear force, wheat bran changed from powdery or granular form to pasty form. In particular, starch was gelatinized and cracked, protein was denatured and recombined and fiber was partially degraded and refined, resulting in a melted state of the bran micro-surface [9,21]. Then, under the strong pressure difference, occurring at the outlet of the extruder, the moisture contained in the bran was instantaneously vaporized and flashed, so that the bran was expanded to form a loose and puffed structure. Other studies also confirmed that the extrusion process was prone to generate the above changes in microstructure and morphology of bran, which was beneficial to be hydrolyzed by enzymes [26]. Such micro-structural deformation should be attributed to the fast and full dissolution of SDF and good hydration features of bran.

### 3.2. Water States of Wheat Dough

Water distribution and mobility in dough have significant influence on the rheological behavior and technical features. Water, either bound to flour ingredients or free, is not homogeneously distributed in dough. To explore the water-migration performance and competitive water sequestration existed between wheat bran and gluten matrix, we used LF-NMR to determine the T_2_ relaxation times and water contents of wheat dough (Figure 3). There are two CPMG proton populations, T_21_ and T_22_, which stand for bound water and free water, respectively. The relaxation time stands for the water mobility and peak area reflects the relative amount of H-bonds. A long relaxation time indicates a high degree of water freedom. For wheat dough, it has been inferred that T_21_ represents the overlapping population of water on the surface of starch granules and water surrounding the gluten strands [27]. According to some researchers, bound water is thought to facilitate gluten formation and contribute to the supramolecular organization of dough structure [15]. T_22_ is the capillary water between gluten network and wheat bran, which is related to the flow properties of the dough [3,15].

As shown in Table 2, with the increase in gluten content, T_21_ relaxation time and A_21_ peak-area ratio were increased. It indicated that the increase of gluten addition increased the amount of water hydrogen-bonded to the polymer matrix while weakening the water affinity. The increased A_21_ peak-area ratio represents the relative content of bound water was increased in dough system. For flour dough with the addition of raw bran (RWD groups), T_21_ relaxation time decreased in company with the increase of A_21_ population and T_22_ relaxation time. Since wheat bran had a stronger water-binding ability than gluten network, the increased A_21_ population in RWD groups possibly contained the water absorbed by wheat bran plus intra-granular water in starch. Furthermore, the left shift of T_21_ peak implied that bran sequestered bound water and prevented its free migration. There was no effect on the free water amount as evidenced from there being no regular changes in A_22_ population. The above observations were in agreement with the fact that wheat bran not only sequestered water but also affected water activity.

For flour dough with the addition of extruded bran (EWD groups), Table 2 shows that T_21_ relaxation times became shorter and A_21_ populations tended to be larger, and T_22_ peaked with the opposite changing trend compared to RWD groups. This phenomenon demonstrated higher populations of more tightly bound water and less populations of free water in EWD groups. As analyzed above, extruded bran contained more SDF with greater hydrophilic groups. Both the exposure of binding sites in bran structure and the damage of starch by extruding treatment should be responsible for the increased A_21_ population. Because of the higher WHC of extruded bran than gluten proteins, water molecules tended to move from free water to bound water as presented by the shortened T_21_ relaxation time. It has been recognized that hydration is supposed to be the important process for protein polymerization and dough formation. Water could act as a solvent that dissolves water-soluble proteins with low molecular weight and induce conformational changes in hydrophobic interactions [15]. Resulting from the partitioning of water between gluten and bran, it is hypothesized that extruded bran sequestered water, causing the partial dehydration of gluten matrix, thus limiting the water participating in the gluten formation process. Furthermore, the amount of extruded bran affected water retention capacity and dough rheology. To verify the hypothesis, the protein secondary structure was characterized for exploring the relationship between water sequestering and gluten development.

### 3.3. Protein Secondary Structure of Wheat Dough

The FTIR spectroscopy was performed to study second structure changes in gluten with the incorporation of wheat bran. The contents of α-helix, *β*-sheet, *β*-turn and random structure were calculated from relative peak areas of 1652~1660 cm^−1^, 1660~1685 cm^−1^, 1620~1644 cm^−1^ and 1644~1652 cm^−1^, respectively [17].

In Figure 4, RWD groups show that *β*-turn and *β*-sheet were major protein secondary structures, which occupied 32.23~39.99% and 28.22~33.26%, respectively, while α-helix and random accounted for 15.63~19.23% and 13.73~16.94%, respectively. The increase of gluten content increased the *β*-turn content with no change in the other three structure contents. As Bock et al. reported, in the model gluten-only dough, gluten predominantly exists as *β*-turn structures. So, with the increment of gluten content in dough samples, the *β*-turn content increased. According to their study, these structures in wheat dough could have a relationship with *β*-spiral domains in glutenin polypeptides. After extrusion treatment, there are many changes in protein secondary structures of gluten in EWD groups. The *β*-turn contents of EWD groups were higher than those of RWD groups, while the *β*-sheet content presented the opposite tendency. With the increase of extruded bran content, *β*-turn content reduced, and *β*-sheet content increased for low-gluten and middle-gluten flour dough. It was suggested that the dramatic increase of *β*-sheet was at the expense of the decreased content of *β*-turn. However, for EWD groups at high-gluten content dough, the conversion of *β*-turn to *β*-sheet was not apparent. Some reports have pointed out that *β*-turn was the preferred structure of gluten in the fully hydrated state [28]. The mechanistic explanation of these empirical observations should be attributed to the redistribution of moisture between extruded bran and gluten in wheat dough as evidenced from the above section. The water-sequestering capability of extruded bran promoted the binding ability of dough to bound water as evidenced by the increased A_21_ and decreased T_21_. The data also showed that the bran could compete with the gluten network for the bound water, which in turn caused protein conformational variations [3,15]. However, the transformation was minimal in higher-gluten dough due to sufficient water being available in the system to adequately hydrate the dough system. The results clearly indicated that extruded bran addition influenced the direct linkage between trans-conformational changes in gluten and gluten-water interaction extent in dough, and the degree of this transformation must be a function of the ratio of extruded bran to gluten content.

In fact, the protein secondary structures are closely associated with the gluten network formation and the gluten strength. According to reports, the *β*-sheet is thought to be the most stable of protein conformation [3], and the high degree of protein polymerization is strongly related to the buildup of the stable protein secondary structures [15]. Therefore, in the lower-gluten dough, the higher proportion of *β*-sheet structure should demonstrate the higher ordered structure and facilitate the polymerization of protein in EWD groups than RWD groups. Addition of extruded bran made the gluten tend to adopt a *β*-sheet configuration, resulting from the water redistribution and gluten dehydration. This structural transformation made the protein aggregate firmly to form a three-dimensional viscoelastic network. The loss of *β*-turn structures and the formation of *β*-sheet structures, thus, implied the formation of a more elastic gluten network. Other studies also showed the reconstruction of protein secondary structures of wheat dough by bran addition. Such a change was considered to be one of the structural elements responsible for dough viscoelasticity [3,21].

### 3.4. Rheological Properties of Wheat Dough

#### 3.4.1. Farinographic Property

In Table 3, water absorption (WA) of all samples showed a gradually increasing trend with the addition of bran, and EWD groups caused a further increase in WA compared to RWD groups. It was typically attributed to the high WHC of extruded sample and its strong water competition capacity. The high presence of hydrophilic groups in SDF, the damage of starch (the starch content in wheat bran was decreased from 33.38 ± 1.37% to 30.01 ± 0.49% after extrusion treatment) and the exposure of binding sites in extruded bran with reduced particle sizes could lead to the higher WA in EWD groups. The results are in agreement with the reports of Gómez et al. [9] and Wang et al. [29], which found that bran addition caused an increase in the of WA of dough. Dough development time (DT) and stability value (ST) stand for the flour strength, with higher values correlating with the stronger doughs [29]. With regard to DT, bran addition raised the DT value for low-gluten dough, which reflected the increased time to reach the maximum dough consistency, and EWD groups showed greater increased tendency. However, with the increment of gluten content, the DT value was reduced with bran addition. With regard to ST, some researchers have found that the addition of bran deteriorated the dough stability, while other studies provided opposite results [9,29]. Our results showed that ST was prolonged for low-gluten flour while it was shortened at higher-gluten conditions with the increase of bran amount. The increased DT and ST in low-gluten flour may be caused by the interference of bran on the gluten formation process caused by the water competition effect, which prolonged the dough-development process and improved the stability. Moreover, the improved stability upon bran addition may also be related to the increased rigidity of dough. For higher-gluten flours, bran possessed a weaker ability to competing for water owing to the less exposed binding sites within a relatively intensive gluten network. This is because bran influenced gluten development by the formation of steric hindrance, or by the dilution of the gluten or by piercing the gas cells, which reduced the gluten connectivity and thus lowered the DT and stabilities. Furthermore, the negative influences were enhanced when extruded bran was added. Possible explanations might be that the powered hydrophilic micro-environment, the liberation of soluble reactive components and the increased particle surface of extruded bran contribute to easier interactions of gluten with reaction components. According to Noort et al. [30], ferulic acid monomer bound to the cell wall could interact with gluten proteins in particular. 

The mixing tolerance index, Ds, was used to assess the softening degree of the dough [3]. The Ds results showed that, in the low-gluten system, raw bran addition improved the hardness of the dough sheet, while the addition of extruded bran further promoted this tendency. Thus, farinographic results suggested that extruded bran under low-gluten conditions had a positive influence on the strength of the gluten network.

#### 3.4.2. Dynamic Rheological Behavior

Figure 5 and Figure 6 show the dynamic rheological properties of dough fortified with 0~25% wheat bran. In all dough samples, both the storage (*G*′) and loss (*G*″) moduli increased with the increasing frequency, which demonstrated their dependence on the frequency [31]. Generally, the dough samples demonstrated a response typical of a cross-linked polymer network mainly manifesting solid-like behavior as evidenced by *G*″ < *G*′ (loss tangent, tan*δ* < 1) [32]. *G*′ and *G*″ values of both RWD and EWD groups showed a rising tendency with the frequency increased from 0.1 to 10 Hz. This behavior has already been reported by many researchers [3,31]. A greater *G*′ normally indicates a more elastic and solid-like structure. As shown in Figure 5 and Figure 6, EWD groups showed stronger mechanical strength than RWD groups. It suggested that the much stiffer doughs had lower flowing tendency. Compared with control dough, RWD groups showed lower tan*δ* values, and EWD groups enlarged the difference. Furthermore, tan*δ* decreased with the increased extruded bran. The increased rheological moduli and reduced loss tangent might be owing to the water competition of wheat bran with the gluten network in the mixing process, which restrained the water uptake of the gluten network and gave rise to interparticle interaction between water and other dough components in a strengthened dough system, finally contributing to its elastic properties. Ronda et al. [33,34] noted that bran prompted solid-like behavior of dough. On this basis, viscoelastic profiles indicated that EWD groups formed a more compact and less extensible gluten network than RWD groups. In the processing of flour products, such as mixing, stretching and fermentation, it is necessary to form a continuous gluten matrix to ensure the stability of the preprepared dough. However, the formation of an excessively compact gluten matrix could make doughs have poor processing adaptability, such as possibility of hard dough.

## 4. Conclusions

The morphological and compositional characteristics, swelling and solvent retention capacity of wheat bran treated by extrusion were analyzed. These experiments confirmed that the solubility of dietary fiber was increased by a factor of 1.47 and the porous structure of bran was formed after extrusion processing. Thus the consequent hydration and swelling capacities were higher in extruded bran. The effect of extruded wheat bran on the structural and rheological properties of wheat dough with different gluten contents was elucidated. The water state and protein secondary structure of gluten dough was studied to reveal the underlying physical mechanism by which bran–gluten interaction impacted dough properties. LF-NMR results revealed that there were two distinct water populations formed in the gluten dough. Addition of extruded bran caused water redistribution, as presented by the larger proportion of tightly bound water to the gluten matrix. The bran–gluten interaction influenced by water sequestering affected the protein secondary structure as evidenced by the conversion of *β*-turn into *β*-sheet especially in lower-gluten dough, which indicated the partial dehydration of gluten and consequently formed a more stable and polymerized gluten network. Farinographic data showed that extruded bran increased DT and ST by factors of 2.25 and 4.82, respectively, and decreased the Ds by 25.81% only at lower-gluten content of dough. Viscoelastic data confirmed that extruded bran increased *G*′, *G*” and reduced tan*δ* of wheat dough at all gluten contents, indicating that the incorporation of extruded bran would improve the dough elasticity. These results would be useful for interpreting the modification effects and mechanism of extrusion on cereal brans and guide the application of the extrusion technique on bran processing and its flour products development.

## Figures and Tables

**Figure 1 foods-12-01813-f001:**
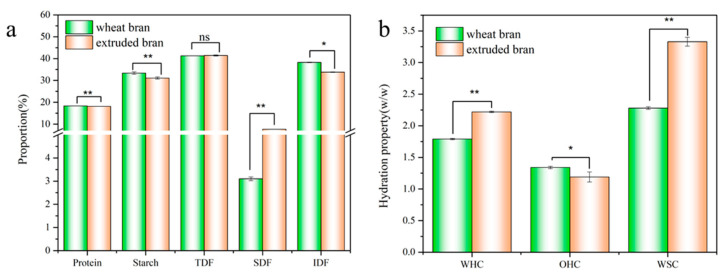
Contents of protein, starch, TDF, SDF and IDF of wheat bran (**a**); WHC, OHC and WSC value (**b**). Compared with raw bran, ns: no significant difference; *: significant difference (*: *p* ≤ 0.05; **: *p* ≤ 0.01).

**Figure 2 foods-12-01813-f002:**
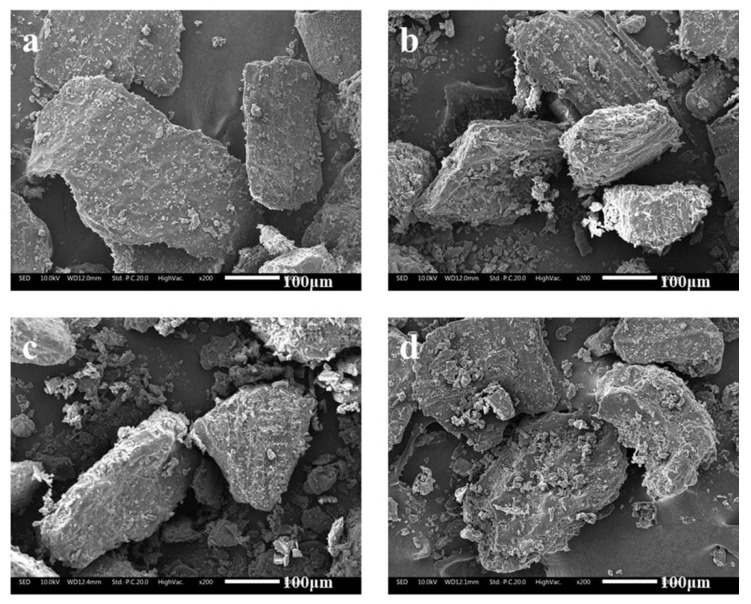
SEM graphs of raw (**a**) and extruded bran (**b**–**d**) powders.

**Figure 3 foods-12-01813-f003:**
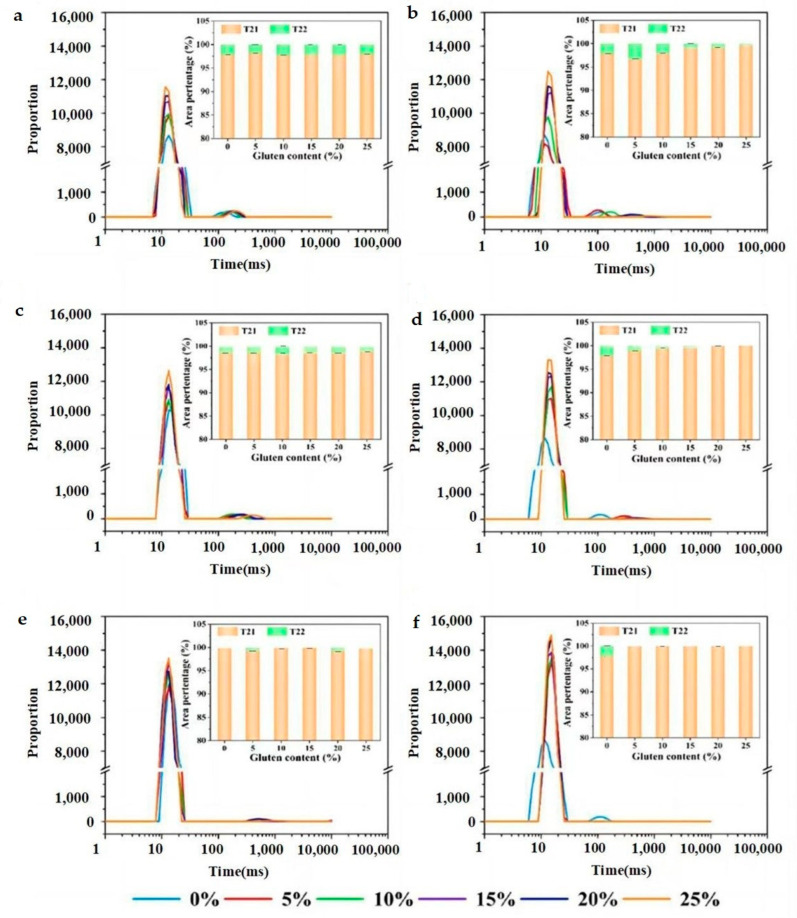
Distributed T_2_ relaxation plots and water content for low- (**a**,**b**), middle- (**c**,**d**) and high- (**e**,**f**) gluten dough with incorporation of raw (**a**,**c**,**e**) and extruded bran (**b**,**d**,**e**).

**Figure 4 foods-12-01813-f004:**
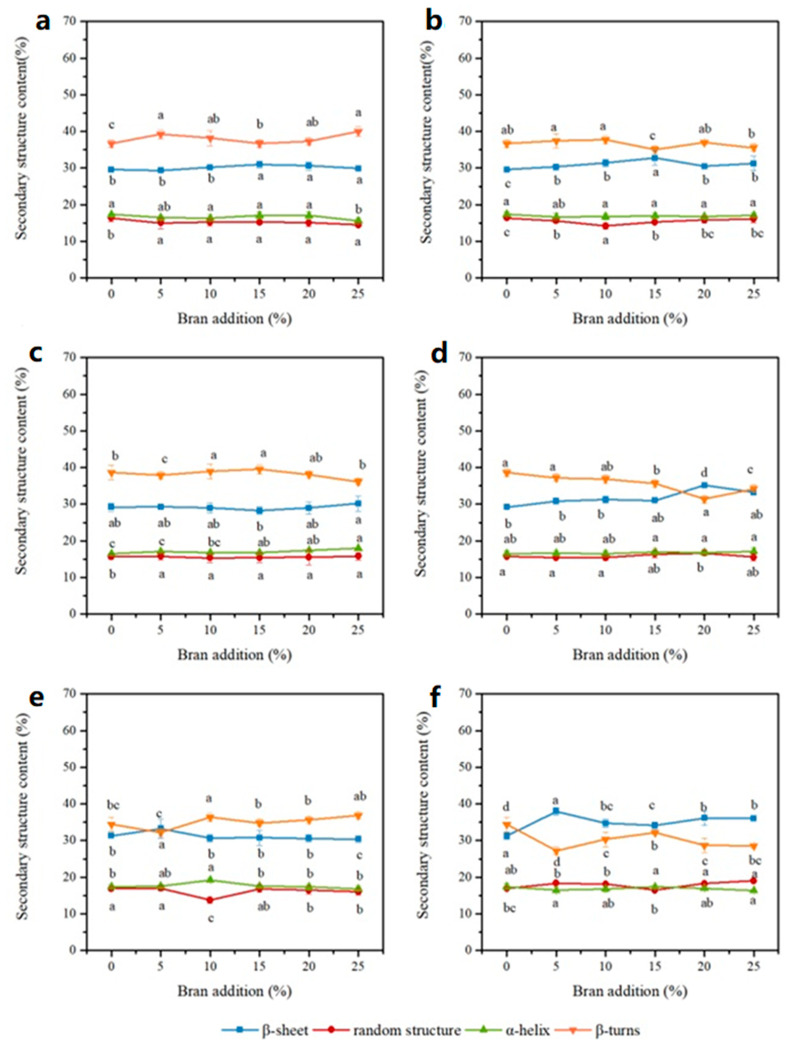
Protein secondary structure contents of wheat dough at high- (**a**,**b**), middle- (**c**,**d**), low- (**e**,**f**) gluten dough with raw (**a**,**c**,**e**) and extruded bran (**b**,**d**,**f**). Dots (mean ± SD, *n* = 3) with different letters have mean values that are significantly different (*p* < 0.05).

**Figure 5 foods-12-01813-f005:**
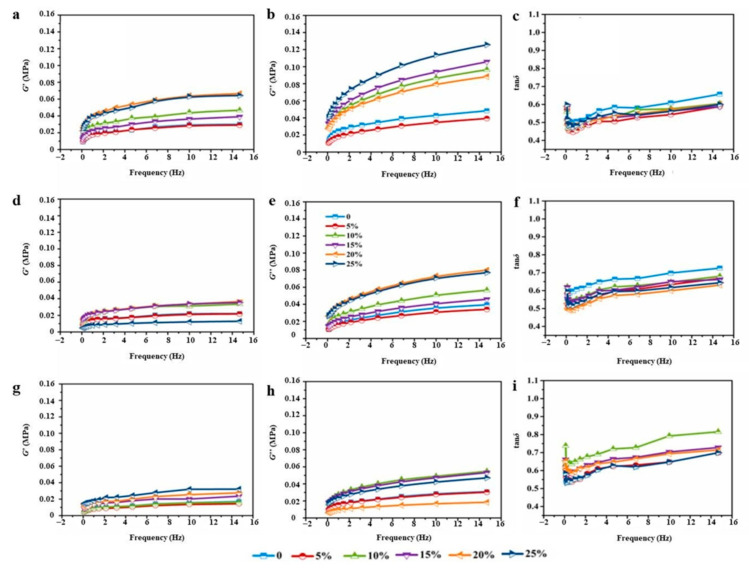
*G*′, *G*″ and tan*δ* of high-gluten (**a**–**c**), middle-gluten (**d**–**f**) and low-gluten (**g**–**i**) dough with raw bran.

**Figure 6 foods-12-01813-f006:**
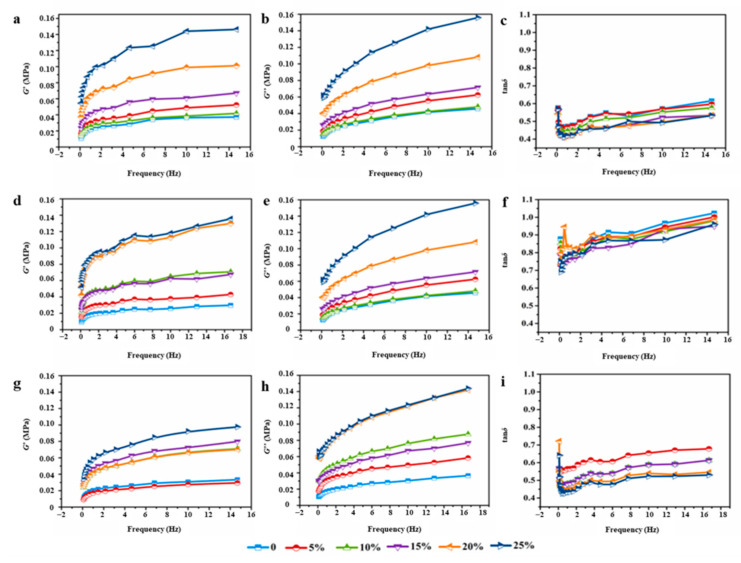
*G*′, *G*″ and tan*δ* of high-gluten (**a**–**c**), middle-gluten (**d**–**f**) and low-gluten (**g**–**i**) dough with extruded bran.

**Table 1 foods-12-01813-t001:** The protein content, wet gluten content and alveograph parameters of wheat flours.

Flour Types	Protein Content/%	Wet Gluten Content/%	Alveograph Parameters		
			*P*/mm	*L*/mm	*W*/(10^−4^ × J)
Low-gluten wheat flour	6.73 ± 0.23	21.56 ± 1.21	54.23 ± 2.36	95.63 ± 2.43	136.92 ± 2.37
Middle-gluten wheat flour	10.48 ± 0.56	26.89 ± 0.58	23.01 ± 0.89	54.99 ± 1.89	89.08 ± 2.02
High-gluten wheat flour	14.83 ± 0.45	34.56 ± 1.37	75.50 ± 2.75	160.88 ± 3.65	213.25 ± 4.03

**Table 2 foods-12-01813-t002:** T_2_ relaxation time and peak-area ratio of wheat dough.

	Bran Addition/%	Raw Bran				Extruded Bran			
		T_21_/ms	T_22_/ms	A_21_/%	A_22_/%	T_21_/ms	T_22_/ms	A_21_/%	A_22_/%
High-gluten flour	0	15.21 ± 0.10 ^a^	618.18 ± 0.00 ^d^	99.87 ± 0.00 ^a^	0.10 ± 0.00 ^d^	−	−	−	−
	5	13.23 ± 0.00 ^b^	444.78 ± 0.00 ^f^	99.07 ± 0.00 ^cd^	0.93 ± 0.00 ^b^	15.11 ± 0.00	−	100.00 ± 0.00	−
	10	13.23 ± 0.00 ^b^	920.05 ± 0.00 ^b^	99.75 ± 0.01 ^ab^	0.17 ± 0.01 ^d^	15.11 ± 0.00	−	100.00 ± 0.00	−
	15	13.23 ± 0.00 ^b^	1344.94 ± 0.39 ^a^	99.82 ± 0.23 ^ab^	0.08 ± 0.02 ^d^	15.11 ± 0.00	−	100.00 ± 0.00	−
	20	12.14 ± 0.55 ^c^	496.63 ± 0.22 ^e^	98.91 ± 0.14 ^d^	1.09 ± 0.01 ^a^	15.11 ± 0.00	−	100.00 ± 0.00	−
	25	13.23 ± 0.00 ^b^	777.22 ± 0.71 ^c^	99.44 ± 0.07 ^bc^	0.55 ± 0.08 ^c^	15.11 ± 0.00	−	100.00 ± 0.00	−
Middle-gluten flour	0	13.22 ± 0.02	214.09 ± 0.00 ^d^	98.80 ± 0.02 ^a^	1.20 ± 0.02 ^e^	−	−	−	−
	5	13.23 ± 0.00	187.51 ± 0.00 ^e^	98.51 ± 0.03 ^b^	1.49 ± 0.03 ^c^	15.11 ± 0.00 ^a^	279.07 ± 0.00 ^c^	98.90 ± 0.01 ^e^	1.09 ± 0.02 ^a^
	10	13.23 ± 0.00	171.99 ± 0.77 ^d^	98.20 ± 0.02 ^d^	1.80 ± 0.02 ^a^	15.11 ± 0.00 ^a^	415.35 ± 0.00 ^b^	99.22 ± 0.05 ^d^	0.78 ± 0.05 ^b^
	15	13.23 ± 0.00	255.98 ± 1.15 ^b^	98.56 ± 0.03 ^b^	1.44 ± 0.03 ^c^	15.11 ± 0.00 ^a^	474.22 ± 2.24 ^b^	99.38 ± 0.05 ^c^	0.62 ± 0.05 ^c^
	20	13.23 ± 0.00	244.43 ± 0.00 ^c^	98.38 ± 0.03 ^c^	1.62 ± 0.03 ^b^	13.23 ± 0.00 ^b^	705.80 ± 1.89 ^a^	99.76 ± 0.03 ^b^	0.18 ± 0.03 ^d^
	25	13.23 ± 0.00	365.92 ± 2.79 ^a^	98.70 ± 0.04 ^a^	1.30 ± 0.03 ^d^	13.23 ± 0.63 ^b^	−	100.00 ± 0.00 ^a^	−
Low-gluten flour	0	13.22 ± 0.02 ^a^	125.99 ± 0.00 ^f^	98.50 ± 0.02	1.50 ± 0.02	−	−	−	−
	5	13.23 ± 0.00 ^a^	157.43 ± 0.68 ^d^	98.50 ± 0.04	1.50 ± 0.04	11.59 ± 0.00	96.65 ± 4.57 ^c^	97.84 ± 0.02 ^d^	2.16 ± 0.02 ^a^
	10	13.23 ± 0.00 ^a^	143.84 ± 0.00 ^e^	98.20 ± 0.04	1.80 ± 0.04	13.23 ± 0.00	164.23 ± 1.76 ^c^	98.43 ± 0.03 ^c^	1.57 ± 0.03 ^b^
	15	12.68 ± 0.55 ^ab^	179.75 ± 0.77 ^c^	98.48 ± 0.26	1.52 ± 0.26	15.11 ± 0.00	415.35 ± 7.21 ^b^	98.96 ± 0.00 ^b^	1.04 ± 0.00 ^c^
	20	12.14 ± 0.55 ^ab^	187.51 ± 0.00 ^b^	98.28 ± 0.03	1.72 ± 0.03	13.23 ± 0.00	415.35 ± 1.96 ^b^	99.02 ± 0.02 ^b^	0.98 ± 0.02 ^c^
	25	11.59 ± 0.00 ^b^	196.37 ± 0.89 ^a^	98.24 ± 0.02	1.76 ± 0.02	13.23 ± 0.00	805.83 ± 5.42 ^a^	99.50 ± 0.13 ^a^	0.50 ± 0.05 ^d^

Values (mean ± SD, *n* = 3) with different letters have mean values that are significantly different (*p* < 0.05).

**Table 3 foods-12-01813-t003:** Farinographic properties of wheat flours.

Flour Type	Bran Addition /%	Untreated Bran				Extruded Bran			
		Water Absorption (WA) /%	Development Time (DT) /min	Stability Time (ST) /min	Degree of Softening (Ds) /FU	Water Absorption (WA) /%	Development Time (DT) /min	Stability Time (ST) /min	Degree of Softening (Ds) /FU
High-gluten flour	0	62.9 ± 0.0 ^f^	15.0 ± 0.1 ^a^	19.6 ± 0.1 ^a^	−	63.1 ± 0.0 ^c^	15.0 ± 0.1 ^a^	19.6 ± 0.0 ^a^	−
	5	64.4 ± 0.1 ^e^	9.1 ± 0.0 ^c^	17.1 ± 0.0 ^b^	−	65.3 ± 0.1 ^c^	13.7 ± 0.2 ^b^	14.1 ± 0.2 ^b^	−
	10	65.6 ± 0.0 ^d^	10.1 ± 0.1 ^b^	15.5 ± 0.1 ^c^	−	68.5 ± 0.1 ^b^	11.1 ± 0.3 ^c^	14.6 ± 0.2 ^b^	−
	15	68.7 ± 0.0 ^c^	9.0 ± 0.0 ^c^	15.7 ± 0.2 ^c^	−	72.7 ± 0.1 ^b^	8.8 ± 0.1 ^d^	11.4 ± 0.4 ^c^	−
	20	70.0 ± 0.2 ^b^	7.2 ± 0.2 ^e^	13.4 ± 0.3 ^d^	−	76.2 ± 0.3 ^a^	8.4 ± 0.1 ^d^	7.8 ±0.2 ^d^	−
	25	74.3 ± 0.1 ^a^	7.8 ± 0.1 ^d^	11.2 ± 0.2 ^e^	−	77.3 ± 0.2 ^a^	7.5 ± 0.2 ^e^	6.4 ± 0.1 ^d^	112.8 ± 7.3
Middle-gluten flour	0	58.8 ± 0.1 ^f^	5.3 ± 0.3 ^a^	12.6 ± 0.2 ^a^	45.0 ± 1.2 ^c^	58.8 ± 0.2 ^e^	5.2 ±0.1 ^a^	13.4 ± 0.1 ^a^	44.4 ± 2.1 ^e^
	5	60.9 ± 0.1 ^e^	3.2 ± 0.1 ^c^	7.8 ± 0.1 ^b^	93.5 ± 2.0 ^b^	62.0 ± 0.0 ^d^	2.5 ± 0.0 ^d^	7.3 ± 0.1 ^b^	72.4 ± 3.2 ^d^
	10	62.8 ± 0.2 ^d^	3.3 ± 0.2 ^c^	6.8 ± 0.1 ^c^	103.2 ± 1.7 ^b^	65.6 ± 0.1 ^c^	3.4 ± 0.2 ^c^	4.8 ± 0.2 ^c^	108.6 ± 9.0 ^c^
	15	65.4 ± 0.1 ^c^	4.2 ± 0.1 ^b^	6.1 ± 0.3 ^d^	116.8 ± 1.5 ^a^	70.0 ± 0.2 ^b^	3.8 ± 0.2 ^b^	4.1 ± 0.1 ^cd^	122.9 ± 6.4 ^bc^
	20	68.7 ± 0.2 ^b^	3.8 ± 0.3 ^b^	5.5 ± 0.2 ^d^	129.1 ± 3.5 ^a^	73.0 ± 0.3 ^a^	4.0 ± 0.0	3.7 ± 0.0 ^d^	136.4 ± 4.8 ^a^
	25	73.1 ±0.1 ^a^	5.3 ± 0.2 ^a^	6.3 ±0.2 ^d^	128.9 ± 2.4 ^a^	75.1 ± 0.1 ^a^	4.0 ± 0.1 ^b^	2.8 ± 0.1 ^e^	135.4 ± 5.1 ^a^
Low-gluten flour	0	49.3 ± 0.1 ^e^	1.2 ± 0.0 ^d^	1.1 ± 0.1 ^e^	132.1 ± 2.1 ^a^	49.3 ± 0.1 ^e^	1.1 ± 0.3 ^d^	1.1 ± 0.3 ^d^	132.1 ± 2.1 ^a^
	5	49.6 ± 0.1 ^e^	1.2 ± 0.0 ^d^	1.4 ± 0.0 ^e^	119.8 ± 4.1 ^b^	49.3 ± 0.0 ^e^	1.3 ± 0.1 ^d^	1.7 ± 0.2 ^d^	126.0 ± 3.3 ^b^
	10	52.5 ± 0.0 ^d^	1.3 ± 0.1 ^cd^	2.7 ± 0.2 ^d^	110.2 ± 1.9 ^c^	52.5 ± 0.1 ^d^	1.5 ± 0.0 ^d^	3.2 ± 0.4 ^c^	121.7 ± 5.0 ^b^
	15	57.4 ± 0.2 ^c^	1.5 ± 0.1 ^d^	3.4 ± 0.1 ^c^	106.6 ± 2.3 ^c^	56.1 ± 0.3 ^c^	3.9 ± 0.2 ^c^	5.4 ± 0.2 ^b^	95.4 ± 2.4 ^cd^
	20	60.7 ± 0.0 ^b^	1.8 ± 0.0 ^b^	5.5 ± 0.2 ^b^	90.3 ± 0.7 ^d^	58.2 ± 0.2 ^bc^	5.3 ± 0.3 ^b^	4.5 ± 0.2 ^bc^	86.2 ± 7.9 ^d^
	25	62.5 ± 0.1 ^a^	3.9 ± 0.1 ^a^	6.4 ± 0.1 ^a^	98.0 ± 1.4 ^d^	63.5 ± 0.1 ^a^	6.7 ± 0.2 ^a^	7.0 ± 0.0 ^a^	45.9 ± 3.7 ^e^

Values (mean ± SD, *n* = 3) with different letters have mean values that are significantly different (*p* < 0.05).

## Data Availability

Data is contained within the article.

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
