# Peer review of "Extrusion Modification of Wheat Bran and Its Effects on Structural and Rheological Properties of Wheat Flour Dough"

_foods, 2023, doi:10.3390/foods12091813_

Round 1

Reviewer 1 Report

Dear authors,

please review the comments in text

Cordially,

Author Response

Dear reviewer, We would like to thank you very much for giving us constructive suggestions which would help us to improve the quality of the paper. We have checked the manuscript and modified it according to your comments. Revised portion are marked in red in the paper. The point-to-point response to the reviewer’s comments is as following:

1.The reviewer’s Comments: Enlarge the Fig. 3.

The authors’ Answer: Thanks for your suggestion. The image pixels and size of Fig. 3 have been increased and enlarged in the revised manuscript.

2. The reviewer’s Comments: Assurance that the data in Table 1 are not duplicated in Fig. 1.

The authors’ Answer: We confirmed that the data in Table 1 are not duplicated in Fig. 1. Fig. 1 demonstrated the chemical composition, swelling capacity and solvent retention capacity of untreated and extruded wheat bran. Table 1 listed the water states in gluten dough with wheat bran addition. Once again, we appreciate for your warm work earnestly, and hope that the corrections will meet with approval. Your sincerely, Ranran Li, Chenyang Wang, Yan Wang, Xuan Xie, Wenjie Sui, Rui Liu, Tao Wu and Min Zhang

Reviewer 2 Report

Dear author,  It is a good effort and written very comprehensively. However there needs to be some changes to be done for improvements like usage of synonyms is poor and inappropriate words are used so try to review and correct them title needs some improvements as it seems extruded induced interaction of bran with gluten, like it feels that both were under extrusion conditions. Improve your title In results and discussion, to the point discussion is needed, few very broad concepts were used that should be specific and precise Composition of bran would help to find out the degree of starch gelatinization, and protein polymerization and so on.. In SEM analysis, if you would describe how much percentage of particles broken down into paste or which component mainly affected.   Other comments are mentioned in the manuscript as well.

Author Response

Dear reviewer, We would like to thank you very much for giving us constructive suggestions which would help us to improve the quality of the paper. We have checked the manuscript and modified it according to your comments. Revised portion are marked in red in the paper. The point-to-point response to the reviewer’s comments is as following:

1.The reviewer’s Comments: It is a good effort and written very comprehensively. However there needs to be some changes to be done for improvements like usage of synonyms is poor and inappropriate words are used so try to review and correct them. Title needs some improvements as it seems extruded induced interaction of bran with gluten, like it feels that both were under extrusion conditions.

The authors’ Answer: Thanks for your good suggestion. Some inappropriate expressions including improper usage of synonyms have been carefully checked and corrected in the revised manuscript. To avoid above misunderstanding, the title has been changed to “Extrusion modification of wheat bran and its effects on structural and rheological properties of wheat flour dough”.

2. The reviewer’s Comments: Improve your title in results and discussion, to the point discussion is needed, few very broad concepts were used that should be specific and precise Composition of bran would help to find out the degree of starch gelatinization, and protein polymerization and so on.

The authors’ Answer: Thanks for pointing it out. The sub-title in Results and Discussion has been improved as below, and some broad concepts have been replaced by the specific and precise descriptions in the revised manuscript. In this manuscript, we characterized and analyzed the main components’ content, microstructure, swelling capacity and solvent retention capacity of extruded wheat bran. And the systematic compositional and structural characterizations of wheat bran treated by extrusion, including the characterization of starch crystalline structure by XRD, the gelatinization degree by DSC and the protein aggregation performance by SDS-PAGE and SLS, are being conducted in experiments and will be analyzed in subsequent articles. 3.1. Physicochemical properties of extruded wheat bran 3.2. Water structure of wheat flour containing wheat bran 3.3. Protein secondary structure of wheat flour dough containing wheat bran 3.4. Rheological properties of wheat flour dough containing wheat bran

3. The reviewer’s Comments: In SEM analysis, if you would describe how much percentage of particles broken down into paste or which component mainly affected.

The authors’ Answer: Thanks for your good comment. Owing to the limited number of SEM micrographs, 3~5 graphs were taken for each sample, which is insufficient to quantitatively reflect the percentage of particles broken down into paste. So we just qualitatively described the changed trend in the microstructure of wheat bran after extrusion treatment. Considering the possible hydrothermal chemical and structural changes that occurred in the main components of bran during extrusion process, the starch gelatinization, protein denaturation and fiber degradation should mainly contribute to the melted state of bran micro-surface, which has been supplemented in the revised manuscript as below. In particular, starch was gelatinized and cracked, protein was denatured and recombined, and fiber was partially degraded and refined, leading to the melted state of bran micro-surface. Once again, we appreciate for your warm work earnestly, and hope that the corrections will meet with approval. Your sincerely, Ranran Li, Chenyang Wang, Yan Wang, Xuan Xie, Wenjie Sui, Rui Liu, Tao Wu and Min Zhang

Reviewer 3 Report

This paper is presenting a study of the influence of extruded wheat bran on the formation of a suitable gluten dough matrix. It explored in detail how bran (raw and extruded) in different proportion modifies the structure of the gluten network for three different gluten content, as well as the rheological properties of the resulting products.  

The paper is overall well written, and the study goals are clearly explained and discussed. I have a few comments related to the statistics, the clarity of some sentences as well as some figures.

The author mentioned that all experiments were performed three times. I find it would be pertinent to include missing error bars on graphs. For example, the rheological study was presented without error bars. How many tests were carried and are the results statistically significant? I have a similar comment for Figure 4 and the inserts of Figure 3. I assume that the main graph in Figure 3 presents just an example of a test performed on a particular sub-sample.

Wording

I found some sentences that probably need some revision:

l65-37 “which interfering.”

l183: “which extremely promoting the conversion…”

l310 “It also forced the bran competed for bound water with the gluten network.”

L433-438 The first sentence is unclear and incomplete. Maybe the full stop should be replaced by a coma… but the last sentence of the paper is extremely unclear. I suggest a revision.

Figures:

Beside from the absence of error bars (or comment on their absence), I noticed a font size problem in Figures 5 and 6. “Frequency” is a bit larger in 5c, 5j, 6c and 6j. For the G’ and G’’ figures, I suggest changing the unit to MPa to reduce the number of 0 on the y-axis.

Author Response

Dear reviewer, We would like to thank you very much for giving us constructive suggestions which would help us to improve the quality of the paper. We have checked the manuscript and modified it according to your comments. Revised portion are marked in red in the paper. The point-to-point response to the reviewer’s comments is as following:

1.The reviewer’s Comments: The author mentioned that all experiments were performed three times. I find it would be pertinent to include missing error bars on graphs. For example, the rheological study was presented without error bars. How many tests were carried and are the results statistically significant? I have a similar comment for Figure 4 and the inserts of Figure 3. I assume that the main graph in Figure 3 presents just an example of a test performed on a particular sub-sample.

The authors’ Answer: Thanks for pointing it out. Unless otherwise specified, three independent trials were performed, with a new batch used for each sample preparation. The error bars have been added on all the graphs. Due to the small values of error in Fig. 3, Fig. 5 (G’ and G’’) and Fig. 6 (G’ and G’’), they are not clearly displayed on the graphs. For example, the data of A21 and A22 in Table 1 correspond to the data in the sub-figure of Fig. 3. In the sub-figure, small errors are unclearly displayed. The statistical analyses of Fig. 1 and Fig. 4 have been supplemented in the revised manuscript.

2. The reviewer’s Comments: I found some sentences that probably need some revision: (1)-L65-37 “which interfering.” (2)-L183: “which extremely promoting the conversion…” (3)-L310 “It also forced the bran competed for bound water with the gluten network.” (4)-L433-438 The first sentence is unclear and incomplete. Maybe the full stop should be replaced by a coma… but the last sentence of the paper is extremely unclear. I suggest a revision.

The authors’ Answer: Thanks for your kind suggestion. All above points have been carefully checked and revised in the manuscript as below. (1) “which interfering...” has been revised to “which would interfere…”. (2) “which extremely promoting the conversion…” has been revised to “which could extremely promote the conversion…”. (3) “It also forced the bran competed for bound water with the gluten network.” has been revised to “The data also showed that the bran could compete with gluten network for the bound water in the dough system.”. (4) These sentences have been revised as below: Viscoelastic data confirmed that extruded bran increased G’, G” and reduced tanδ of wheat dough at all gluten contents, indicating that the incorporation of extruded bran would strengthen the dough elasticity. These results would be useful to interpret the modification effects and mechanism of extrusion on cereal’s brans and further guide the application of extrusion technique on bran processing and its flour products development.

3. The reviewer’s Comments: Beside from the absence of error bars (or comment on their absence), I noticed a font size problem in Figures 5 and 6. “Frequency” is a bit larger in 5c, 5j, 6c and 6j. For the G’ and G’’ figures, I suggest changing the unit to MPa to reduce the number of 0 on the y-axis.

The authors’ Answer: Thanks for pointing it out. We are very sorry for our careless check about the axis titles. The font size of axis labels has been adjusted in Fig. 5 and Fig. 6, and the unit of y-axis has been changed to MPa. Once again, we appreciate for your warm work earnestly, and hope that the corrections will meet with approval. Your sincerely, Ranran Li, Chenyang Wang, Yan Wang, Xuan Xie, Wenjie Sui, Rui Liu, Tao Wu and Min Zhang

Reviewer 4 Report

The manuscript provides a molecular insights on the effect of extruded bran on dough properties. Although the topics is of interest for the scientific community, in some parts it was difficult to follow the focus on the study. Since the role of bran in dough was widely addressed in previous reports, I would suggest to rewrite the discussion pointing out on the effect of extrusion process rather than on the effect of bran level or type of gluten. Moreover, the novelty of the study should be better highlighted since this is not the first time a study deals with the extrusion of bran. From a methodological stand point, I have some concerns about the water level in dough for water mobility and protein secondary structure analysis. Lsst but not least, te relation of dough properties with product features would have added value to the study. In other words, it is not clear if the observed change will have an impact on dough handling (including stickiness) and bread quality. Further comments in the attached file.

Author Response

Dear reviewer, We would like to thank you very much for giving us constructive suggestions which would help us to improve the quality of the paper. We have checked the manuscript and modified it according to your comments. Revised portion are marked in red in the paper. The point-to-point response to the reviewer’s comments is in the attachment file.

Dear reviewer,

We would like to thank you very much for giving us constructive suggestions which would help us to improve the quality of the paper. We have checked the manuscript and modified it according to your comments. Revised portion are marked in red in the paper. The point-to-point response to the reviewer’s comments is as following:

  1. The reviewer’s Comments: Since the role of bran in dough was widely addressed in previous reports, I would suggest to rewrite the discussion pointing out on the effect of extrusion process rather than on the effect of bran level or type of gluten. Moreover, the novelty of the study should be better highlighted since this is not the first time a study deals with the extrusion of bran.

The authors’ Answer: Thanks for your good suggestion. We have rewritten the Results and Discussion section, focusing on analyzing the changes in structural and rheological properties of dough caused by the extrusion treatment of wheat bran. And the novelty of the study has been highlighted in Conclusion section in the revised manuscript. The innovation of this work mainly lies in the effect of extruded wheat bran on the structural and rheological properties of wheat dough with different gluten contents, so as to provide theoretical guidance for the development of different types of wheat flour with bran addition.

  1. The reviewer’s Comments: From a methodological stand point, I have some concerns about the water level in dough for water mobility and protein secondary structure analysis.

The authors’ Answer: We are sorry for the unclear description on the dough preparation method. Wheat dough was prepared by mixing flour (high-gluten flour, middle-gluten flour, low-gluten flour) and wheat bran (0%, 5%, 10%, 15%, 20%, 25%) with water until the consistency of the dough peaked in a Brabender farinograph using a 50 g mixing bowl. Doughs were collected at the end of mixing and sealed with cling film standing about 30 minutes to wake up. The LF-NMR characterization was performed within 30~45 min. And the dough was freeze-dried to conduct the FT-IR characterization. This method has been added in the revised manuscript.

  1. The reviewer’s Comments: The relation of dough properties with product features would have added value to the study. In other words, it is not clear if the observed change will have an impact on dough handling (including stickiness) and bread quality.

The authors’ Answer: Thanks for your kind suggestion. In the bakery industry, a better understanding of the rheological properties of flour dough during processing is significant, due to the relationships between these properties and quality attributes of the final products. In the mixing, stretching and fermentation process of flour products, it is necessary to form the continuous gluten matrix to ensure the stability of the preprepared dough. The formation of an excessively compact gluten matrix could make doughs have poor processing adaptability, such as possibility of hard dough. And the reduction of dough elasticity by disturbing the formation of gluten network would lead to the interrupted gas cells with subsequent low bread volume and poor baking quality. Above information on the relationship between rheological properties and dough handling has been described in the revised manuscript.

  1. The reviewer’s Comments and authors’ Answer: Introduction

Q1: The introduction lacks of mentioning the most recent findings on the same topic: https://doi.org/10.3390/foods9060738; https://doi.org/10.1016/j.jcs.2022.103577; https://doi.org/10.1080/87559129.2022.2097689; https://doi.org/10.1016/j.lwt.2021.112543.

A: Thanks for pointing it out. The most recent references you mentioned have been added in the revised manuscript as below:

  1. Roye, C.; Henrion, M.; Chanvrier, H.; De Roeck, K.; De Bondt, Y.; Liberloo, I.; King, R.; Courtin, C.M. Extrusion-cooking modifies physicochemical and nutrition-related properties of wheat bran. Foods 2020, 9, 738.
  2. Zhang, S.; Jia, X.; Xu, L.; Xue, Y.; Pan, Q.; Shen, W.; Wang, Z. Effect of extrusion and semi-solid enzymatic hydrolysis modifications on the quality of wheat bran and steamed bread containing bran. J. Cereal Sci. 2022, 108, 103577.
  3. Khanpit, V. V.; Tajane, S. P.; Mandavgane, S. A. Extrusion for soluble dietary fiber concentrate: critical overview on effect of process parameters on physicochemical, nutritional, and biological properties. Food Res. Int. 2022, 1-22.
  4. Wu, Y.; Ye, G.; Li, X.; Wang, L.; Liu, Y.; Tan, B.; Shen, W.; Zhou, J. Comparison of quality characteristics of six reconstituted whole wheat flour with different modified bran. LWT Food Sci. Technol. 2022, 153, 112543.
  5. The reviewer’s Comments and authors’ Answer: Materials and Methods

Q2: What is harvested, not bran. Please modify the sentence.

A: This sentence has been modified as “Wheat bran was purchased from Xiangyuan Jinxiangzhai Trading Co., Ltd., Shanxi Province, China.”.

Q3: If the wheat selection was based on gluten content, please include the amount of gluten of each sample. Besides gluten/protein content, the quality of gluten should be mentioned. Thus, please include the alveograph W, P and L values for all the samples.

A: The wheat flour was selected based on gluten contents. The protein content and wet gluten content of low-gluten, middle-gluten and high-gluten wheat flour were determined to be 6.73% and 21.56%, 10.48% and 26.89%, 14.83% and 34.56%, respectively. Alveograph characteristics of wheat flour using Chopin Alveograph (Model Alveographe NG, Chopin, France) were determined using AACC methods (2000). The protein content, wet gluten content and alveograph parameters of wheat flours were listed in Table 1 in the revised manuscript.

Q4: what about the particle size of the raw bran? How many batches were produced? This is important for assessing the process repeatability.

A: The raw wheat bran is in the form of thin slices with a diameter of 3~5 mm and a thickness of 0.5~1 mm. The wheat bran was obtained from a small-scale flour processing plant in Shanxi, which would produce 50000 tons of wheat flour and around 8800 tons of wheat bran annually.

Q5: Reference 28 does not report the details to repeat the experiments. Not clear why potassiunm bromide was used. The method reported by [20] should be carried out.

A: The correct reference has been added in this part in the revised manuscript as below:

  1. Bock, J.E.; Damodaran, S. Bran-induced changes in water structure and gluten conformation in model gluten dough studied by Fourier transform infrared spectroscopy. Food Hydrocoll. 2013, 31, 146-155.

The principle of potassium bromide pellet technique is to mix the sample with potassium bromide and use high pressure to press them into thin sheets. Because of its good transparency and chemical stability, potassium bromide is suitable as a carrier for tested samples. The thin sheets have good transparency and can greatly reduce light scattering and absorption. In this way, the molecular structure of dough samples can be analyzed accurately. The dough sample preparation method for FTIR characterization has been added in the revised manuscript as below:

A freeze-dried dough sample (1 mg) and potassium bromide (150 mg) were ground into a fine powder in an agate mortar, under infrared irradiation. The mixture was then pressed into a plate and the FTIR spectrum were recorded by 40 scans in the range of 400~4000 cm−1 with a resolution of 4 cm−1.

  1. The reviewer’s Comments and authors’ Answer: Results and Discussion

Q6: what about lipid content?

A: The lipid content of raw wheat bran was 4.08±0.24%, which has been added in the revised manuscript.

Q7: please, include reference here

A: The relevant reference has been cited in the revised manuscript.

  1. Menis-Henrique, M. E. C.; Scarton, M.; Piran, M. V. F.; Clerici, M. T. P. S. Cereal fiber: Extrusion modifications for food industry. Curr. Opin. Food Sci. 2020, 33, 141-148.

Q8: is this difference significant?

A: This difference between the starch content of raw bran and the starch content of extruded bran is significant with p ≤0.01.

Q9: please clarify it

A: Extrusion has been reported to cause the hydration and denaturation of proteins, which in turn affected their biological activity or potency, such as the improvement of intrinsic digestibility, the reduction of trypsin inhibitor activity and the drops in serum and liver cholesterol levels, according to the following reference. These specific examples have been described in the revised manuscript.

  1. Korhonen, H.; Pihlanto-Leppäla, A.; Rantamäki, P.; Tupasela, T. Impact of processing on bioactive proteins and peptides. Trends Food Sci. Tech. 1998, 9(8-9), 307-319.

Q10: Please, include statistical analysis (t-test)

A: We have done the statistical analysis of Fig. 1 in the revised manuscript.

Q11: These results are not original.

A: Indeed, several studies have reported the impact of extrusion treatment on the SDF content of wheat bran and its swelling capacity and solvent retention capacity (Ref.[]). However, the innovation of this work mainly lies in the effect of extruded wheat bran on the structural and rheological properties of wheat dough with different gluten contents, so as to provide theoretical guidance for the development of different types of wheat flour with bran addition.

Q12: please clarify it

A: According to some literatures, the desirable hydration performance could facilitate water absorption of dietary fibers and their expansion to a gelation form, thereby increasing the food viscosity (like ice cream) and delaying or preventing the excess cholesterol absorption in food. Additionally, the great hydrophilic property could promote bowel movements and thus effectively prevent constipation, diverticulosis and colon cancer. Above explanation has been supplemented in the revised manuscript.

  1. Akalın, A. S.; Kesenkas, H. A. R. U. N.; Dinkci, N. A. Y. I. L.; Unal, G. Ü. L. F. E. M.; Ozer, E. L. İ. F.; Kınık, O. Enrichment of probiotic ice cream with different dietary fibers: Structural characteristics and culture viability. J. Dairy Sci. 2018, 101(1), 37-46.
  2. Wang, C.; Song, R.; Wei, S.; Wang, W.; Li, F.; Tang, X.; Li, N. Modification of insoluble dietary fiber from ginger residue through enzymatic treatments to improve its bioactive properties. LWT Food Sci. Technol. 2020, 125, 109220.
  3. Yan, J.; Hu, J.; Yang, R.; Zhang, Z.; Zhao, W. Innovative nanofibrillated cellulose from rice straw as dietary fiber for enhanced health benefits prepared by a green and scale production method. ACS Sustainable Chem. Eng. 2018, 6(3), 3481-3492.

Q13: are these differences significant?

A: This difference of WHC, WSC and OHC of raw bran and extruded bran are significant with p ≤0.01, p ≤0.05, p ≤0.01 as shown in Fig. 1.

Q14: Please make a comment on the effect/impact of type of flour.

A: Effect of flour type on water structure of wheat flour has been analyzed in the revised manuscript as below:

The data in Table 1 showed that with the increase in gluten content, T21 relaxation time and A21 peak area ratio were increased. It indicated that the increment of gluten addition increased the amount of water hydrogen-bonded to polymer matrix while weaken the water affinity. The increased A21 peak area ratio represents the relative content of bound water was increased in the dough system. There was no effect on free water amount as evidenced from no regular changes in A22 population during the entire water migration process.

Q15: Please, specify the meaning of "technical" properties

A: The dietary fiber has a strong water retention capacity since it has a larger number of hydroxyl groups in the molecular structure. This brings a better chance for the dietary fiber to occupy the available water molecules in the dough, resulting in a retardation for gluten network development and a slower process for gluten hydration. As such, the modifications in the dough rheology and bread quality for a whole wheat formula are induced by the interactions of gluten proteins, dietary fiber and water molecules. The addition of bran (dietary fiber) in the dough resulted in the bakery products with a smaller loaf volume, higher moisture content and firmer crumb texture, due to the negative technical effects of dietary fiber on the baking performance of dough.

  1. Sun, X.; Wu, S.; Koksel, F.; Xie, M.; Fang, Y. Effects of ingredient and processing conditions on the rheological properties of whole wheat flour dough during breadmaking-A review. Food Hydrocolloid. 2023, 135, 108123.

Q16: please include reference here

A: The relevant reference has been cited in the revised manuscript as below:

  1. Lu, Z.; Seetharaman, K. 1H Nuclear magnetic resonance (NMR) and differential scanning calorimetry (DSC) studies of water mobility in dough systems containing barley flour. Cereal Chem. 2013, 90(2), 120–126.

Q17: what is the meaning of increased peak area?

A: In this sentence, this refers to the increase of peak area ratio, which means the ratio of A21 peak area to the total peak area. The peak area of T2 represents the relative content of hydrogen protons and the water absorption by hydrophobic components. The higher ratio of A21 peak area, the more water exists as bound water in the dough system. The explanation has been added in the revised manuscript.

Q18: Please point out the effect of extrusion rather than the effect on bran addition or gluten quantity.

A: The effect of extrusion has been pointed out in the revised manuscript as below:

After extrusion treatment, there are many changes in secondary structures of gluten in EWD groups. The β-turn content of EWD groups were higher than that of RWD groups, and the β-sheet content presented the opposite tendency. As extruded bran content increased from 0 to 25%, β-turn content decreased and β-sheet content increased for low-gluten and middle-gluten flour dough.

The results clearly indicated that extruded bran addition influenced the direct linkage between trans-conformational changes in gluten and gluten-water interaction extent in dough. And the degree of this transformation must be a function of the ratio of extruded bran to gluten content.

Q19: please add a comment based on previous findings.

A: The discussion on the effect of gluten content on protein secondary structures has been added in this section as below:

As Bock et al. reported, in the model gluten only dough, gluten predominantly exists as β-turn structures. So with the increment of gluten content in dough samples, the β-turn content increased.

According to their study, these structures in wheat dough might be related to the β-spiral domains in glutenin polypeptides.

Q20: Please add reference for this statement.

A: The relevant reference has been cited in the revised manuscript as below.

  1. Xiong, L.C.; Zhang, B.J.; Niu, M.; Zhao, S.M. Protein polymerization and water mobility in whole-wheat dough influenced by bran particle size distribution. LWT Food Sci. Technol. 2017, 82, 396-403.
  2. Sui, W.J.; Xie, X.; Liu, R.; Wu, T.; Zhang, M. Effect of wheat bran modification by steam explosion on structural characteristics and rheological properties of wheat flour dough. Food Hydrocoll. 2018, 84, 571-580. 469.

Q21: please include data

A: The data has been added in this sentence in the revised manuscript as below:

(the starch content in wheat bran was decreased from 33.38±1.37% to 31.10±0.49% after extrusion treatment)

Q22: not clear, please rephrase it

A: This sentence has been rewritten as below:

However, with the increment of gluten content, the DT value was reduced with bran addition.

Q23: how do the authors explain the increase in "stability" upon the bran addition?

A: With the addition of wheat bran, the increased stability of low-gluten flour may be contributed wheat interfered with the gluten development by its water competition effect, which promoted the connectivity of gluten network, prolonged dough development process and strengthened the stability. This explanation has been added in the revised manuscript.

Q24: how this information can be related to dough handling?

A: In the bakery industry, a better understanding of the rheological properties of flour dough during processing is significant, due to the relationships between these properties and quality attributes of the final products. In the mixing, stretching and fermentation process of flour products, it is necessary to form the continuous gluten matrix to ensure the stability of the preprepared dough. The formation of an excessively compact gluten matrix could make doughs have poor processing adaptability, such as possibility of hard dough. And the reduction of dough elasticity by disturbing the formation of gluten network would lead to the interrupted gas cells with subsequent low bread volume and poor baking quality. Above information on the relationship between rheological properties and dough handling has been described in the revised manuscript.

    Once again, we appreciate for your warm work earnestly, and hope that the corrections will meet with approval.

    Your sincerely,

    Ranran Li, Chenyang Wang, Yan Wang, Xuan Xie, Wenjie Sui, Rui Liu, Tao Wu and Min Zhang

Round 2

Reviewer 4 Report

Dear Authors,

the manuscript has been revised and the previous comments addressed.

Two more considerations:

- the novelty of the study (in comparison to the current body of literature) should be included in the Introduction section

- I suppose that the increase in dough "stability" upon bran addition to low gluten flour is related to dough "rigidity" rather to gluten strenghenening.

Kind regards

Author Response

Dear reviewer, We would like to thank you for giving us constructive suggestions which would help us to improve the quality of the paper. We have checked the manuscript and modified it according to the reviewer’s comments. Revised portion are marked in blue in the paper. The point-to-point response to the reviewer’s comments is as following: 1. The reviewer’s Comments: The novelty of the study (in comparison to the current body of literature) should be included in the Introduction section. The authors’ Answer: Thanks for your kind suggestion. The novelty of this work has been supplemented to the Introduction section in the revised manuscript as below. Previous literatures showed that extrusion could change the physico-chemical properties of wheat bran and prompt the bioavailability and functionality of soluble dietary fiber (SDF) in bran materials [1,4,14]. But the specific influencing mechanism of treated bran on the structure and rheology of wheat dough still remained to be further unveiled. Therefore, it is necessary to explore the interactions of extruded bran and gluten protein in dough system and on these bases evaluate the influence of extruded bran on structural and rheological properties of wheat dough, for the aim of providing valuable guidance for applying extrusion as an effective modification technology on the commercial production of bran-containing flour products. The novelty of this work is to reveal the interaction variations of extruded bran and gluten protein in dough system with different gluten levels and further explore the effects of extruded bran on the structural and rheological properties of wheat dough. 2. The reviewer’s Comments: I suppose that the increase in dough "stability" upon bran addition to low gluten flour is related to dough "rigidity" rather to gluten strengthening. The authors’ Answer: Thanks for pointing it out. This opinion has been added in Results and Discussion section (3.4.1 Farinographic property) as below. Besides, the improved stability upon bran addition may also be related to the increased rigidity of dough. Once again, we appreciate for your and all the reviewers’ warm work earnestly, and hope that the corrections will meet with approval. Your sincerely, Ranran Li, Chenyang Wang, Yan Wang, Xuan Xie, Wenjie Sui, Rui Liu, Tao Wu and Min Zhang